# Anticoagulation Management during Extracorporeal Membrane Oxygenation—A Mini-Review

**DOI:** 10.3390/medicina58121783

**Published:** 2022-12-03

**Authors:** Filip Burša, Peter Sklienka, Michal Frelich, Ondřej Jor, Tereza Ekrtová, Jan Máca

**Affiliations:** 1Department of Anesthesiology and Intensive Care Medicine, University Hospital of Ostrava, 708 00 Ostrava-Poruba, Czech Republic; 2Department of Intensive Care Medicine and Forensic Studies, Faculty of Medicine, University of Ostrava, 708 00 Ostrava-Poruba, Czech Republic; 3Institute of Physiology and Pathophysiology, Faculty of Medicine, University of Ostrava, 708 00 Ostrava-Poruba, Czech Republic

**Keywords:** extracorporeal membrane oxygenation, anticoagulation, heparin, argatroban, bivalirudin, antiIIa, antiXa, aPTT

## Abstract

Extracorporeal membrane oxygenation (ECMO) has been established as a life-saving technique for patients with the most severe forms of respiratory or cardiac failure. It can, however, be associated with severe complications. Anticoagulation therapy is required to prevent ECMO circuit thrombosis. It is, however, associated with an increased risk of hemocoagulation disorders. Thus, safe anticoagulation is a cornerstone of ECMO therapy. The most frequently used anticoagulant is unfractionated heparin, which can, however, cause significant adverse effects. Novel drugs (e.g., argatroban and bivalirudin) may be superior to heparin in the better predictability of their effects, functioning independently of antithrombin, inhibiting thrombin bound to fibrin, and eliminating heparin-induced thrombocytopenia. It is also necessary to keep in mind that hemocoagulation tests are not specific, and their results, used for setting up the dosage, can be biased by many factors. The knowledge of the advantages and disadvantages of particular drugs, limitations of particular tests, and individualization are cornerstones of prevention against critical events, such as life-threatening bleeding or acute oxygenator failure followed by life-threatening hypoxemia and hemodynamic deterioration. This paper describes the effects of anticoagulant drugs used in ECMO and their monitoring, highlighting specific conditions and factors that might influence coagulation and anticoagulation measurements.

## 1. Introduction

Extracorporeal membrane oxygenation (ECMO) is a highly specialized method used as a life-saving technique for patients with the most severe forms of respiratory or cardiac failure. The invasiveness of ECMO creates a high risk of potentially fatal complications. Disorders of hemocoagulation during ECMO therapy are common, particularly due to the contact of the patient’s blood with thrombogenic materials in the extracorporeal circuit. Thus, anticoagulation therapy is required to prevent ECMO circuit (especially oxygenator) thrombosis. The anticoagulant effects should be easily adjustable in response to specific hemocoagulation tests. It is, however, necessary to keep in mind that the standard tests used in daily practice for the monitoring and setup of the anticoagulation treatment may be unsuitable for the monitoring of anticoagulation status when using specific anticoagulants and when such an unsuitable test is used; it may provide inaccurate results resulting in a further increase in the risk of bleeding or thromboembolism. Moreover, critically ill patients on ECMO frequently exhibit conditions accompanied by coagulopathy, including sepsis, trauma, thromboembolic disease, or various hereditary predispositions for coagulopathy [1,2]. Mansour et.al. analyzed bleeding and thrombotic complications in venovenous (VV) ECMO patients and found that bleeding events were associated with increased in-hospital mortality (adjusted OR 2.91; 95% CI 1.94–4.4), unlike thrombotic events (adjusted OR 1.02; 95% CI 0.68–1.53) [3]. Similarly, Stokes et al. reported that bleeding may be a more serious event than thrombosis and, in view of this fact, lower anticoagulation targets should be considered, especially in VV ECMO patients (bleeding event had worse in-hospital survival compared with nonbleeding (*p* = 0.02), while thromboembolic events were not associated with survival (*p* = 0.73)) [4]. Some authors have shown that lower or even no anticoagulation could be safe and feasible [5,6]. 

When speaking of ECMO anticoagulation, one must keep in mind that the goals are different from full anticoagulation such as in cardiac surgery. Here, the goal is to maintain the ECMO circuit free of thrombi while not preventing physiological coagulation processes in injured organs. Anticoagulation in critically ill patients on ECMO must, therefore, be carefully and individually titrated.

Although novel, more biocompatible surface materials for ECMO circuits are being developed, anticoagulation therapy remains a routine part of ECMO treatment [7]. The ideal anticoagulant for ECMO should (1) be intravenously deliverable; (2) have a short elimination half-life with a fast-fading effect, or a specific antidote, or be reversible in its effects; (3) have an excellent safety profile, with minimal risk of allergic reactions, no toxic metabolic products, etc.; (4) have predictable interactions with the ECMO circuit and patient; (5) have predictable effects on the coagulation system; and (6) allow precise monitoring of its effects and be easily adjusted to maintain its concentration in a safe range. 

Balancing anticoagulation to be effective without complications is a challenging task, and the ideal range of anticoagulation doses can be very narrow. None of the available anticoagulation tests can assess all parts of the coagulation cascade and the overall effect of the drugs on fibrin formation within the body. Blocking an individual coagulation factor (as measured by the appropriate test and target values) does not necessarily warrant that a thrombus will not form. On the other hand, a below-target result of an appropriate test (and, thus, probably a low effect of the drug) does not rule out bleeding, which could arise from another disorder within the coagulation system (rather than from blockage of the monitored factor). Moreover, it is impossible to simulate in a laboratory all interactions with biological surfaces that are essential for hemostasis, such as collagen and endothelial cells. Obviously, the clinical goal is an uncomplicated ECMO run, not a strong blockade of a particular coagulation factor or the blood level of the anticoagulant. In addition, the anticoagulant effect must be interpreted in the context of the total levels of thrombin and fibrin (hemostatic capacity) produced prior to the initiation of anticoagulant therapy. We usually monitor the intensity of individual factor blockade, but we do not know the amount of thrombin that we have to block. 

The hemostatic potential is fundamentally influenced by platelets (both their total number and function/dysfunction) and by natural anticoagulation factors (antithrombin, heparinoids, endothelial interactions, etc.). The risks of thrombosis or bleeding specific to the patient’s clinical situation should always be taken into account. 

The 2021 ELSO (Extracorporeal Life Support Organization) anticoagulation guidelines [8] show that treatment decisions about anticoagulation should be highly individualized. There is no uniform anticoagulation management regarding agents, monitoring, and therapeutic targets [9]. In view of this fact, we aimed to provide an overview of the anticoagulants that may be used during ECMO therapy, highlighting the pros and cons of individual anticoagulants and comparing the newer anticoagulation drugs to those widely used for ECMO treatment, such as heparin. We do not discuss antiplatelet therapy during ECMO here, as this treatment may be continued in patients who have an additional indication for antiplatelet therapy. Moreover, as monitoring of the anticoagulation effects is crucial during ECMO, a brief overview of the monitoring methods for the use of ECMO anticoagulation is also provided here. 

### 1.1. Anticoagulants Used in ECMO

Below, we provide a detailed overview of the anticoagulants used in ECMO. A brief summary is also provided in Table 1.

### 1.2. Unfractionated Heparin

#### General Characteristics and Dosage

Unfractionated heparin **(UFH)** is the primary anticoagulant used in ECMO because of the familiarity with the use, price, and the possibility of reversal by antidote [10]. Heparin-like substances are physiologically present in the organism as an endogenous sulfated polysaccharide bound as a component of the heparans lining the inner walls of the vascular system. These substances are formed in the liver, gut, and lungs, and their molecular weight ranges from 15 to 19 kDa [11]. For anticoagulation purposes, pharmaceutical (unfractionated as well as fractionated, see below) heparin is supplemented into the organism. The primary role of UFH lies in maintaining hemostatic homeostasis through the regulation of binding of the plasmatic glycoprotein antithrombin (AT) to Factors II and Xa. 

AT plays a key role in the coagulation balance and links coagulation and inflammation [12,13]. This plasma glycoprotein is produced by the liver, with a normal plasma concentration of 1.5 to 2.0 mg/L. Together with proteins C and S, it belongs to the group of the main endogenous anticoagulants responsible for the physiological inhibition of the clotting system. Its anticoagulant potential consists primarily of the irreversible inhibition of thrombin by binding to the active site of the protein, but it also inhibits fXa and, to a lesser degree, also Factors IXa, XIa, and XII, tissue plasminogen activator, plasmin, and kallikrein. Furthermore, AT can bind to heparin and glycosaminoglycans containing a specific receptor [14]. The normally low anticoagulation activity of AT is amplified several thousand times by UFH. AT activity <41% in septic patients was associated with a significantly poorer outcome compared to those with AT above 71%; AT normalization also had an effect on patient mortality [15]. However, supranormal AT activity did not benefit critically ill patients [16,17] and increased the risk of bleeding complications [18]. Moreover, the immunomodulating and antimicrobial effects of AT are also inhibited when heparin is used, which might negatively affect the inflammatory respons [16]

UFH also blocks fXa, fIXa, fXIa, the tissue factor/Factor VIIa complex, and activates the tissue factor pathway inhibitor. Moreover, it is involved (UFH, heparan sulfate, and more than 400 similar substances) in angiogenesis and in the regulation of inflammation [19]. In addition, heparin is involved in the signaling pathways of growth factors [20]. and has antitumor effects [21]. Heparin also has certain antiviral and antibacterial effects [22]; direct virucidal effects on some other viruses have been also reported [21]. UFH, however, increases bone resorption, and its long-term administration causes osteoporosis [23]. As mentioned above, the anticoagulant activity of heparin is primarily determined by its ability to inhibit coagulation factors through AT. It is cleared from the body by the rapid metabolism in endothelial cells and, to a lesser degree, by renal clearance. 

ELSO recommends starting with a UFH bolus of approximately 50–100 units/kg before cannulation, followed by the infusion at 7.5–20 units/kg/h [8].

### 1.3. Advantages and Limitations

Heparin does not inactivate thrombin already bound to fibrin; this thrombin, therefore, remains active and clot growth can continue despite heparin therapy. The inability of UFH to inactivate fibrin-bound thrombin, and the high degree of UFH binding to plasma proteins, endothelial cells, and macrophages results in the relatively poor predictability of the UFH effect. UFH is also ineffective in inhibiting the platelet-bound Factor Xa and phospholipid-bound Factor Va-Xa complex. In critically ill patients, the plasma levels of acute-phase proteins are elevated, and the binding of heparin to these proteins can further reduce the effect of UFH [24]. On the other hand, in the later phases of illness, when the levels of acute-phase proteins decrease, free heparin blood levels can grow through its release from these acute-phase proteins, which can cause a sudden UFH overdose with a risk of severe bleeding. 

The availability of a heparin antagonist, protamine sulphate, which can completely reverse the anticoagulant effect of heparin, is an advantage of UFH. Protamine is a protein derived from fish sperm with the ability to bind to heparin and produce its inactive salt. One milligram of protamine sulphate neutralizes approximately 100 units of heparin. Because the half-life of heparin is 60–90 min when given as an intravenous infusion, the total amount administered over the previous 2–2.5 h should be taken into account when calculating the protamine sulphate dose to completely antagonize UFH. 

Heparin use is associated with two risks that need to be discussed, namely, heparin-induced thrombocytopenia (HIT) and heparin resistance (HR).

**Heparin-induced thrombocytopenia** can be classified into Type I, which is a nonantibody-mediated reaction to UFH, and Type II, which is antibody-mediated. HIT II is more severe, and it is caused by the heparin-induced formation of IgG antibodies that bind to the complex heparin-platelet Factor 4 (PF4). These immunocomplexes lead to abnormal and irreversible platelet activation and aggregation and can result in severe arterial and venous thrombotic complications and/or thrombocytopenia [25]. HIT usually develops a few days after the initiation of UFH therapy but may occur as much as weeks after the cessation of heparin therapy. Scoring systems predicting HIT have been developed (Warkentin 4T score, Lilo-Le Louet score, the heparin-induced thrombocytopenia expert probability score, etc.); their utility is, however, limited [26,27]. Patients at risk should be screened for HIT with an anti-PF4 heparin antibody test, which has good sensitivity but is limited by low specificity. Hence, positive results of this test should be verified by the serotonin release assay [26,27]. In the study by Marter et al., the absolute risk of HIT with UFH and low-molecular-weight heparin therapy were 2.6% and 0.2%, respectively [28]. However, the incidence of HIT in patients on ECMO can reach as much as 17% [29,30]. The risk factors for HIT development in ECMO patients include 5 days or more of UFH use, post-surgical ECMO (cardiac surgery), and female gender [31]. If HIT is suspected or confirmed, it is necessary to discontinue UFH and switch to another type of anticoagulant.

**Heparin resistance (HR)** is a phenomenon observed when the UFH standard dosage does not yield the expected values in the appropriate test (i.e., the activated partial thromboplastin time, aPTT; the activated clotting time, ACT; or the anti–Factor Xa test) and doses need to be increased [32]. In practice, it is best recognizable in situations when the patient reacts to low doses of UFH, but after reaching a certain threshold (which can differ interindividually), no or only a minimal increase in the test results is observed. This phenomenon is caused by the nonspecific binding of UFH to plasma proteins, high fibrinogen, and Factor V levels, or, most frequently, by low levels of AT. AT deficiency could be acquired or, rarely, congenital. Acquired AT deficiency is caused by reduced AT synthesis (hepatic dysfunction), increased clearance (nephropathy), or due to the patient’s underlying illness [33]. According to the literature, HR affects approx. 5.4–26% of patients undergoing cardiopulmonary bypass [34]. Novelli et al. refer to HR in SARS-CoV-2–infected patients, 75% of whom had heparin resistance to some degree [35]. Administering AT (in the case of low AT levels) without changing the UFH infusion rate can significantly increase the anticoagulation effect of heparin, with an increased risk of bleeding. If HR occurs, changing UFH for another drug or very careful monitoring of the UFH effect is recommended.

### 1.4. Key Points

UFH remains the anticoagulant of choice in many ECMO centers due to the fact of its easy titration, low cost, familiarity with use, and antidote availability. 

In ECMO patients, a UFH initial bolus of approximately 50–100 units/kg before cannulation followed by the infusion at 7.5–20 units/kg/h is recommended.

UFH is usually titrated to aPTT; this parameter, however, evaluates the activity of many coagulation factors (i.e., not only the UFH effect, see Table 2).

HIT, heparin resistance, ineffectiveness against pre-existing clots, and binding to acute-phase proteins could complicate UFH use.

## 2. Low-Molecular-Weight Heparin

### 2.1. General Characteristics

Low-molecular-weight heparin (LMWH) has been clinically used for more than 40 years. In some clinical situations, it might be considered an alternative to UFH during ECMO support [36]. LMWH consists of fragments of heparin produced by enzymatic heparin depolymerization. Several structurally distinct LMWHs with anticoagulant activity are known, with a molecular weight generally between 4 and 5 kDa. Compared with UFH, LMWHs have longer plasma half-lives and improved bioavailability, and their dose-dependent effect is better predicted [36]. They also show a greater inhibition of fXa than of fIIa (higher anti-Xa/anti-IIa ratio). The ability of LMWHs to inactivate thrombin is only 25–50% of that of UFH; thus, their anticoagulant activity acts predominantly on fXa. LMWH nonspecifically binds to macrophages, endothelial cells, platelets, osteoblasts, and other plasma proteins. This prolongs the plasma half-life of LMWH but also reduces the risk of heparin-induced thrombocytopenia (HIT) due to the decrease in binding to PF4. LMWHs, however, cannot be completely antagonized with protamine. 

LMWHs are popular anticoagulation drugs in the ICU and appear to also have potential in treating ECMO patients. Gratz et. al. reported the use of LMWH in the perioperative management of lung transplant patients on ECMO [36]. There was no difference in severe bleeding complications between LMWH and UFH patients; the LMWH group, however, had significantly fewer thromboembolic events (20% vs. 50%, *p* = 0.01).

We do not have as strong evidence regarding LMWH pleiotropic effects against infections and immunomodulation as we do with UFH [37]. LMWH was, however, shown to inhibit the enzymatic activity of heparinase [38,39], inhibit the viral spread [40], and reduce vascular leakage and inflammation [37].

To the best of our knowledge, there are no official recommendations regarding LMWH dosage in ECMO. In the aforementioned study, Gratz et al. [36] used a standard LMWH dosage for their ECMO patients.

### 2.2. Advantages and Limitations

The use of LMWH with ECMO is very limited, especially due to the fact of its prolonged half-life compared to UFH, which can make dose adjustment to a target range difficult. Moreover, protamine cannot antagonize LMWH. On the other hand, the dose-dependent effect of LMWH is more predictable; a direct method of monitoring (antiXa, see Table 2) is available, and LMWH has a better safety profile compared to UFH. 

### 2.3. Key Points

LMWHs have longer plasma half-lives and improved bioavailability compared to UFH, and their dose-dependent effect is more predictable; anticoagulant activity acts predominantly on fXa:

No antidote;

Reduced risk of HIT;

Direct method of monitoring (see Table 2) by antiXa;

Difficult titration to target levels due to the long plasma half-lives;

So far limited use in ECMO.

## 3. Ultra Low-Molecular-Weight Heparins

### 3.1. General Characteristics

Ultra low-molecular-weight heparins (ULMWHs), such as fondaparinux, have a pentasaccharide structure similar to those of UFH and LMWH. Their molecular weight is usually approximately 1.5 to 3.5 kDa. Their bioavailability is better and plasma half-life is longer than in the case of LMWH; the bleeding risk is also lower [41]. They block only fX and have no anti-IIa effects. ULMWHs do not bind to PF4 and do not cause HIT. ULMWHs can, therefore, be used as an alternative to UFH in HIT, and their use in ECMO has been, although rarely, reported previously [42].

### 3.2. Advantages and Limitations

Due to the fact of their longer half-life and consequent longer effect duration, ULMWHs may be unsuitable in situations potentially requiring a rapid change in the anticoagulation, which limits their use in ECMO.

### 3.3. Key Points

ULMWHs could be used as alternatives to UFH in HIT, but long half-lives make them difficult to use in ECMO

## 4. Direct Thrombin Inhibitors

### 4.1. General Information

Direct thrombin inhibitors (DTIs) bind directly to thrombin, and their effect is independent of any cofactors (such as AT) [43]. They inhibit both soluble thrombin and fibrin-bound thrombin. Only a small amount is bound to plasma proteins, and DTIs do not affect platelets. Thus, their effect is more predictable compared with UFH [44]. Representatives of DTIs include lepirudin and desirudin, which are not widely used in critically ill patients and ECMO (longer half-life, irreversible complexes with thrombin, and risk of severe anaphylaxis) [45]. Dabigatran is an oral anticoagulant only, which makes it unsuitable for ECMO. However, two other representatives of DTIs, bivalirudin and argatroban, are suitable for ECMO.

### 4.2. Advantages and Limitations

The main advantages of bivalirudin and argatroban over heparin lie in their ability to inhibit thrombin bound to fibrin, function independent of AT, eliminate HIT risk, the possibility of the direct monitoring of their effect by anti-IIa monitoring (see below), and noninterference with acute-phase proteins. They do not have antidotes, but this is not a major problem as their half-lives are short. Improved steerability and predictability of their effects also constitute important advantages [46]. Nevertheless, large, prospective randomized trials are needed to confirm the superiority of DTIs before they can be considered the first-line anticoagulant choice in ECMO.

### 4.3. Bivalirudin

#### 4.3.1. General Characteristics

Bivalirudin binds to the thrombin catalytic site and anion-binding exosite I in a concentration-dependent manner [47]. The fact that the binding is reversible may reduce the risk of severe bleeding [47]. This drug is cleared both by proteolytic cleavage in plasma and renal mechanisms, predominantly glomerular filtration. It is, therefore, unsuitable for use in patients with renal failure [48]. The onset of its effect is very quick, with a short half-life of 25 min. Bivalirudin also shows activity against thrombin-mediated platelet activation [49].

There are much data on its administration during percutaneous coronary intervention [50] and ECMO [30,51] including induction boluses and dosage during ECMO [52,53]. Bolus doses in ECMO start between 0.4 and 0.5 mg/kg [54,55](only limited data from case reports are available), followed by continuous infusion at rates of 0.05 to 0.5 mg/kg/h. Bivalirudin could be used with no initial bolus with continuous infusion, which varied very widely from 0.028 to 0.5 mg/kg/h [53,55], to achieve the required activated clotting time (ACT) of approximately 200 s. It is necessary to keep in mind that the dosage must be lowered in patients with renal dysfunction [56]. Patients on renal replacement therapy required insignificantly higher doses compared with those without such treatment (0.041 vs. 0.028 mg/kg/h; *p* = 0.2) [57]. The ACT targets are usually 180–220 s and aPTT 45–88 s [49]. There is limited information on the use of bivalirudin as a standard anticoagulation on ECMO; the available data, however, suggest it to be safe [33].

Limited data suggest that bivalirudin may possess certain anti-inflammatory effects [58].

#### 4.3.2. Advantages and Limitations 

Bivalirudin appears to be generally safe in ECMO, with risks comparable to UFH [59], but some papers refer to an increased rate of bleeding complications compared to UFH on ECMO patients [53]. Bivalirudin partially dissociates from the catalytic site of thrombin, making this free thrombin available for reaction with fibrinogen, especially under low blood flow, which may complicate ECMO. In the low-flow regions of the ECMO circuit (e.g., oxygenator) or inside the heart, regional bivalirudin effect could decrease below critical levels (due to the fact of its very short half-life and dissociation from thrombin), which may result in thrombus formation [60].

Accumulation of bivalirudin in patients with renal failure can lead to bleeding. Raucini et al. reported a “heparin-like effect” (i.e., significantly shorter R-times on thromboelastography in tests with vs. without heparinase) [61]. Patients in whom this heparin-like effect was observed (more than half of those in the study) were more likely to have sepsis (30% vs. 5.6%), which could result from the release of heparinoids from the glycocalyx and mast cells. 

#### 4.3.3. Key Points

Reversible binding to thrombin, very short half-life, and no antidotes;

Inappropriate in patients with renal failure due to the fact of renal clearance, acceptable in HIT or HR;

It is suitable also in ECMO initiation (with a bolus initial dose followed by continuous infusion);

Possibility of direct monitoring by antiIIa and standard monitoring with aPTT (see below);

Can cause the formation of thrombi in regions with low blood flow.

### 4.4. Argatroban

#### 4.4.1. General Characteristics

Argatroban is another DTI that has been used for many years, especially in HIT patients [62], and it seems to be a good choice for the general ECMO population. It binds reversibly to the catalytic site of both free thrombin and fibrin-bound thrombin in the coagulum. Its effect is approximately linearly dose-dependent, and its short circulating half-life of approximately 40 min makes it easy to control [63]. It does not react with UFH-induced antibodies. Argatroban is eliminated by the liver; renal dysfunction or renal replacement therapy, therefore, do not influence its dosage. Hepatopathy is not a contraindication, but doses have to be reduced, especially if drugs affecting CYP3A4 are administered concurrently [64]. Very high doses of argatroban may interfere with fibrinogen measurement kits and may lead to falsely low fibrinogen readings [65]. Argatroban is more easily bound to proteins (20% to albumin and 34% to alpha-1-acid glycoprotein) than other DTIs. In addition, its levels may be affected by lidocaine administration (binding to alpha-1-acid glycoprotein) [66]. Moreover, argatroban may enhance reperfusion in patients with myocardial infarction who received tissue plasminogen activator [67].

Additionally, argatroban was reported to have an antiviral activity [68].

The initial bolus (100–200 µg/kg) of argatroban may be accompanied by significant bleeding; initiating therapy with continuous doses of approx. 0.2 µg/kg/min, therefore, appears to be a better choice [69]. The initial argatroban dose recommended by the manufacturer is 2 µg/kg/min, but most papers report a 10 times lower initial dose for ECMO patients [69]. In patients with severe hepatic dysfunction, a further reduction of the dosage to as little as 0.02 µg/kg/min is advisable [70]. In moderate hepatopathy (overall bilirubin above 25 µmol/L), it is reasonable to reduce the dose to less than 0.25 µg/kg/min (compensating for a two- to three-fold prolongation of half-life) to an aPTT target of 45–60 s [71]. Dingman et al. reported an inverse correlation between the needed argatroban dosage and illness severity according to the modified SOFA (Sequential Organ Failure Assessment) score [72]. On the other hand, Beiderlinden et al., who investigated argatroban dosage and severity of liver dysfunction by indocyanine green clearance (a marker of hepatic perfusion), found no such correlation or necessity to adjust the argatroban doses [73].

For ECMO, apart from the indications in HIT, there is still no clear evidence favoring argatroban to UFH as a first-line anticoagulant [74,75]. In terms of safety, there are no data suggesting it has a worse safety profile than UFH, and bleeding complications are low [75]. aPTT is the most frequently used monitoring mode, with wide target ranges [76]; some authors also report the use of ACT [77]. In summary, most studies appear to target aPTT of 45 or 50–60 or 70 s, with a target point around 50 s and an infusion rate of 0.1–0.3 µg/kg/min. Bleeding is reported to occur mainly when aPTT exceeds 75 s, and thromboembolic complication mainly when aPTT falls below 50 s [75].

#### 4.4.2. Advantages and Limitations

The resistance to DTIs’ anticoagulation effect (i.e., the need to increase doses to reach target values, in this case usually aPTT) has not been as extensively studied as UFH resistance and, where ECMO support is concerned, is limited to case reports [78,79]. The mechanism of the resistance is not precisely known, but it could be associated with hyperfibrinogenemia or high levels of fVIII. In high doses, the relationship between the argatroban dosage and aPTT may become nonlinear and, hence, careful titration is necessary [78,79].

Argatroban appears to be the most promising candidate for replacing UFH as the first-choice drug thanks to its better effect predictability, the possibility of direct effect monitoring (see Table 2), and no risk of HIT or HR; moreover, it also acts against fibrin-bound thrombin and does not interfere with acute-phase proteins. It seems appropriate to reduce its dose in critical care patients, whose hepatic function is often altered. The advantages of argatroban over bivalirudin include its independence from renal dysfunction and the lack of thrombotic events associated with low blood flow. The data describing argatroban use in ECMO are limited, but the available reports seem to support its safety and efficacy [33].

#### 4.4.3. Key Points

Promising candidate for replacing UFH as the first-choice drug in ECMO (not confirmed);

Binds reversibly to the catalytic site of both free thrombin and fibrin-bound thrombin in the coagulum;

Suitable for patients with renal failure, acceptable in HIT or HR;

In hepatopathy, dosage should be reduced;

Acceptably short half-life, and no problems in low-flow regions;

No antidotes;

Possibility of the direct monitoring by antiIIa and standard monitoring with aPTT (see below);

Starting ECMO with boluses can lead to bleeding, and continuous infusion is more appropriate.

## 5. ECMO Anticoagulation Monitoring

Anticoagulation monitoring is, of course, crucial for controlling the optimum anticoagulation level (not only) in ECMO patients. However, although there are many methods for measuring the effectiveness of anticoagulation therapy, no single method can be considered a universally applicable, optimal method [8]. Standard laboratory tests usually evaluate only a certain part of the coagulation cascade. In effect, controlling anticoagulation treatment on the basis of a single test may fail to reflect the overall hemostasis status and, therefore, increase the risk of thrombotic or bleeding complications. For example, normal aPTT values rule out neither bleeding nor the need for stronger anticoagulant activity. In addition, the laboratory coagulation tests evaluate the ability of blood to clot in vitro, which can differ from clotting in vivo. Hence, anticoagulation measurements in ECMO patients should include the intensity of inhibition of individual coagulation factors (aPTT, antiXa, anti IIa, etc.), global measurements of thrombin and fibrin formation (viscoelastic methods), and the clinical effects of anticoagulation (ECMO run) [8].

Although proper monitoring constitutes an inherent part of ECMO anticoagulation treatment, providing a full detailed overview of the monitoring methods would excessively inflate this paper. Still, not to omit this crucial topic, we summarize the most important features of the individual anticoagulation monitoring methods in Table 2.

## 6. Future Directions

Despite the manufacturer’s recommendations on the dosage of individual anticoagulants or even guidelines by expert societies [8], the principle of anticoagulant-spare approach (i.e., the initial use of several times lower doses than recommended and their careful titration to achieve proper anticoagulation level (considering appropriate tests)) appears to be a promising alternative [80,81]. Where anticoagulation is contraindicated (such as intracranial hemorrhage or other life-threatening bleeding events), even anticoagulant-free ECMO can be considered; in such cases, however, extreme care must be taken, and special attention must be paid to the oxygenator thrombosis risk assessment. So far, however, the evidence discussing this approach is limited [82,83,84].

Another way to effect ECMO anticoagulation management could be represented by anticoagulants targeting novel factors such as XI and XII; although only animal data are available so far, this direction of research appears promising [85,86].

Improving the biocompatibility of ECMO tubing to reduce the risk of thrombotic events is another promising direction of research. New circuit materials with surface passivation (phosphorylcholine, albumin, and poly-2-methoxyethylacrylate), biomimetic surfaces (heparin, nitric oxide, and direct thrombin inhibitors), and/or endothelization are under investigation and can further reduce the risk of thrombotic events and, in effect, reduce the need for anticoagulation therapy during ECMO [7].

Most importantly, prospective randomized controlled trials comparing the use of UFH and novel anticoagulants, such as argatroban and bivalirudin, need to be performed to provide sufficient evidence supporting a potential change in clinical practice. Increasing evidence suggests that these novel anticoagulants could be preferable to the traditionally used UFH but, at present, such evidence is not sufficient to justify the routine use of argatroban/bivalirudin in everyday clinical ECMO practice [46,87,88,89,90,91,92,93,94,95].

## 7. Conclusions

The clinical goal of anticoagulation in ECMO is to maintain blood flow and prevent thrombus formation without bleeding. This requires careful individualized titration of patients’ anticoagulation status, which is also associated with the use of multiple anticoagulation tests, as each of those provides a different piece of information and these tests are not interchangeable. Hence, anticoagulation measurements in ECMO patients should include the intensity of the inhibition of individual coagulation factors (aPTT, antiXa, anti IIa, etc.), global measurements of thrombin and fibrin formation (viscoelastic methods), and the clinical effects of anticoagulation (ECMO run). 

Although UFH is still the most widely used anticoagulant in ECMO therapy, novel anticoagulants, such as argatroban or bivalirudin, appear to be very good alternatives but need to be confirmed by prospective randomized controlled trials. 

## Figures and Tables

**Table 1 medicina-58-01783-t001:** Anticoagulants most commonly used in ECMO and their main characteristics. LMWH-low molecular weight heparin;UFH-unfractionated heparin;AT-antithrombin;DTIs-direct thrombin inhibitors;HIT-heparin induced thrombocytopenia; aPTT- activated parcial tromboplastine time.

	Advantages	Disadvantages	Standard Monitoring Test	Doses	Target
**Unfractionated heparin**	-familiar use, price -antagonisation by protamine	-indirect effect through antithrombin, heparin resistance -risk of HIT -less predictable effect -ineffective against pre-existing clots	aPTT antiXa	start 50–100 units/kg, continuous 7.5–20 units/kg/h	50–70 0.3–0.7
**LMWH**	-dose-dependent effect is better predicted than UFH -AT independency	-long half-life -no direct antagonisation	antiXa	no specific ECMO recommendation	0.5–1.0
**Fondaparinux**	-use in HIT -AT independency, blocks purely fX	-longer plasma half-life than LMWH -no direct antagonisation	antiXa	no specific ECMO recommendation	0.5–0.7
**DTIs**	-need no co-factors for their effect (no AT) -inhibit both soluble thrombin and fibrin-bound thrombin -do not affect platelets, use in HIT	-price -nonlinear relationship between dosage and aPTT in high doses (DTIs resistance)			
**Bivalirudin**	-bolus doses in ECMO start -rapid regulation due to the short half-life	-inappropriate in renal failure -risk of trombus formation in low blood flow regions	aPTT antiIIa	start 0.4 and 0.5 mg/kg, continuous 0.05 to 0.5 mg/kg/h	45–88 -------
**Argatroban**	-appropriate half-life /well controlled, well prevents formation of thrombi in low flow -independent on renal failure	-risk of bleeding after bolus doses -dose reduction in hepatopathy	aPTT antiIIa	0.25–0.02 µg/kg/min	45–60 0.4–0.7

**Table 2 medicina-58-01783-t002:** Main properties of anticoagulation tests for ECMO. PT-prothrombin time; INR-international normalised ratio; ACT-activated clotting time; TT-thrombin time; FDP-Fibrin degradation products; NOAC-novel oral anticoagulants.

	Advantages	Disadvantages	Drug to Be Monitored	Normal Values	Target Values
**ACT**	-bedside test from full blood -originally designed for heparin -fast and easy to use	-influenced, besides the effect of heparin, also by many other factors (thrombocytopenia, thrombocytopathy, decreased levels of coagulation factors, hypothermia, anemia, and hypofibrinogenemia) -does not measure the clot strength -ACT values below <200 s do not correspond linearly to heparin activity -poor correlation with the effect and dose of heparin, especially in terms of bleeding complications	heparin	70–90 s	180–200 s
**PT/INR**	-originally designed for coumarin derivatives (e.g. Warfarin)	-not useful for routine ECMO monitoring -susceptible to bias (prolongation: antiphospholipid antibodies, elevated fibrin degradation products, polycythemia, liver disease, vitamin K deficiency; shortening: prolonged transport of a blood sample to the laboratory for analysis)	coumarin derivatives	10–13 s/0.8–1.2	unsuitable for ECMO
**aPTT**	-originally designed for heparin -price -familiar and wide use	-inaccurate if levels of fibrinogen and fVIII are low -nonlinear relationship with DTIs dosage -affected by a number of factors (prolongation: hemodilution, high level of fibrin degradation products, vitamin K deficiency, polycythemia, lupus anticoagulant; shortening: complicated venepuncture) -based on the assumption that the patient’s baseline aPTT is comparable to normal controls. In critically ill patients, baseline aPTT often differs from normal controls, confounding the target values for anticoagulation	heparin DTIs	30–40 s	45–60 s 50–70 s
**TT**	-could be used to distinguish between UFH and DTIs effect when switching between drugs (TT is very sensitive to the presence of DTIs)	-affected by fibrinogen dysfunction or deficiency, presence of paraprotein or high FDP levels	heparin, DTIs	less than 20 s	unsuitable for ECMO
**Viscoelastic tests**	-assess global hemostasis -exclude effect of UFH (eventual NOAC) -assess fibrinolysis	-unknown relevance and target ranges for individual parameters in ECMO			
**antiXa activity**	-directly measures the activity of fXa blockade (determines the degree of inhibition of fXa)	-adequacy of antiXa may fail to preclude thrombus formation -distorted by high bilirubin in liver failure and free hemoglobin -not ideal as an isolated parameter in ECMO -does not consider multiple other factors influencing coagulation (e.g., hypofibrinogenemia or thrombopathy)	LMWH	0 U/mL	0.3 to 0.7 UI/mL
**antiIIa activity**	-directly measures the activity of fIIa blockade (determines the degree of inhibition of fIIa) -drug-calibrated method for DTIs	-adequacy of antiIIa may not preclude thrombus formation -higher doses of argatroban result in flattening of the concentration–response relationship, and overdose is more difficult to determine -not ideal as an isolated parameter in ECMO	DTIs	0 U/mL	0.4 to 1.5 µg/mL

## Data Availability

Not applicable.

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
