# Peer review of "Anticoagulation Management during Extracorporeal Membrane Oxygenation—A Mini-Review"

_medicina, 2022, doi:10.3390/medicina58121783_

Round 1

Reviewer 1 Report

In their manuscript the authors provided a comprehensive review of the available data concerning the appropriate use and monitoring of anticoagulant therapy during extracorporeal membrane oxygenation. Challenges in administration of anticoagulants in this context are well presented along with future perspectives regarding better alternatives to unfractionated heparin. 

When discussing about unfractionated heparin, we suggest to make a more clear distinction between the endogenous heparin and the pharmaceutical product. Also, using capital letter (F) for coagulation factors is recommended.

Author Response

Response to the Reviewer No 1

Comments and Suggestions for Authors

In their manuscript the authors provided a comprehensive review of the available data concerning the appropriate use and monitoring of anticoagulant therapy during extracorporeal membrane oxygenation. Challenges in administration of anticoagulants in this context are well presented along with future perspectives regarding better alternatives to unfractionated heparin. 

When discussing about unfractionated heparin, we suggest to make a more clear distinction between the endogenous heparin and the pharmaceutical product. Also, using capital letter (F) for coagulation factors is recommended.

Thank you for your kind evaluation of our work.

We acknowledge that the discussion on UFH might have been confusing as the described mechanism relates to both endogenous and artificially supplemented compounds. We have amended the first paragraph to read:

Unfractionated heparin is the primary anticoagulant used in ECMO because of the familiarity with use, price, and the possibility of reversal by antidote10. Heparin-like sub-stances are physiologically present in the organism as an endogenous sulfated polysac-charide bound as a component of the heparans lining the inner walls of the vascular sys-tem. These substances are formed in the liver, gut, and lungs and their molecular weight ranges from 15 to 19 kDa. For anticoagulation purposes, pharmaceutical (unfractionated as well as fractionated, see below) heparin is supplemented into the organism. The primary role of UFH lies in maintaining hemostatic homeostasis through the regulation of binding of the plasmatic glycoprotein antithrombin (AT) to Factors II and Xa.

We have also amended the capitalization where speaking of individual coagulation factors as suggested, thank you for noticing this.

Reviewer 2 Report

Dear authors,

Thank you sharing your manuscript. In this review the authors, attempted to review the anticoagulation management among ECMO patients. Overall, very well written with appropriate literature evidence. 

I have the following comments to make: 

In the introduction:

-You may modify this sentence as " anticoagulation therapy is required to prevent ECMO circuit including oxygenator thrombosis". Also change this in the abstract.

- I don't think the authors need to mention the COVID-19 ECMO studies. Please delete Mansour et al & Stokes et al 

In the anticoagulants used in ECMO section:

-Table 1, for UFH the target anti Xa should be 0.3-0.7 and for LMWH 0.5-1

-Please mention any dosage cutoff for UFH to suspect heparin resistance

- In the LMWH section, please delete "To the best of our knowledge, there are no official recommendations regarding LMWHs dosage in ECMO" and mention the ELSO guideline.

- Please describe the current literature evidence of heparin therapy monitoring, apTT vs. anti Xa vs. aPTT + anti Xa

- In the table 2, ACT disadvantage column 1st line with typo error, please edit

- In the Bivalirudin section, please delete " To the best of our knowledge, the pleiotropic effect against viruses etc. observed in UFH has not been confirmed in bivalirudin. Limited data suggest that bivalirudin may possess certain anti-inflammatory effects."

- Throughout the manuscript, authors cited COVID-19 literature. I don't think this is needed, given the topic is broadly on anticoagulation in EMCO pts, not pertinent to a specific cohort. So, please delete. 

Author Response

Response to the Reviewer No 2

Dear authors,

Thank you sharing your manuscript. In this review the authors, attempted to review the anticoagulation management among ECMO patients. Overall, very well written with appropriate literature evidence. 

Thank you very much for your kind words.

I have the following comments to make: 

In the introduction:

-You may modify this sentence as " anticoagulation therapy is required to prevent ECMO circuit including oxygenator thrombosis". Also change this in the abstract.

Thank you for noticing this, you are right that thrombosis prevention relates to the entire circuit, not just to the oxygenator. We have amended the sentence in the abstract as follows:

Anticoagulation therapy is required to prevent ECMO circuit thrombosis. It is, however, associated with an increased risk of hemocoagulation disorders.

In the Introduction, we have changed it to read:

Thus, anticoagulation therapy is required to prevent ECMO circuit (especially oxygenator) thrombosis.

- I don't think the authors need to mention the COVID-19 ECMO studies. Please delete Mansour et al & Stokes et al 

It is true that the mention of COVID-19 in the text is unnecessary; however, both these studies (only Mansour was COVID-related) are mentioned to convey the message that bleeding events are more critical than thrombotic events. Both these studies are in agreement, which also underlines the fact that much of what has been reported for COVID-19 patients is valid generally for ECMO. For this reason, we would prefer to keep them there, we have just deleted the mention of SARS-CoV-2. The paragraph now reads as follows:

Mansour et.al. analyzed bleeding and thrombotic complications in venovenous (VV) ECMO patients and found that bleeding events were associated with increased in-hospital mortality (adjusted OR 2.91;95% CI 1.94-4.4), unlike thrombotic events (adjusted OR 1.02;95% CI 0.68-1.53). Similarly, Stokes et.al. reported that bleeding may be a more serious event than thrombosis and, in view of this fact, lower anticoagulation targets should be considered, especially in VV ECMO patients.

In the anticoagulants used in ECMO section:

-Table 1, for UFH the target anti Xa should be 0.3-0.7 and for LMWH 0.5-1

Thank you for noticing, amended as requested.

-Please mention any dosage cutoff for UFH to suspect heparin resistance

It is difficult to give a universal cut-off that would indicate heparin resistance. There are papers stating such cut-offs (e.g. 35,000 units/day by Durrani et al.) but in our opinion, the key for determining heparin resistance lies rather in the dynamics of reaction, i.e., the obvious loss of reactivity with a further additional increase in the dose. In some individuals, this cutoff may arise at lower levels UFH doses, in others at higher ones and establishing some universal cutoffs could be rather misleading and contradict the current trend of personalized medicine. Nevertheless, we agree that it is necessary to highlight the clinical relevance and, for this reason, we have added the following sentence:

In practice, it is best recognizable in situations when the patient reacts to low doses of UFH but after reaching a certain threshold (which can differ interindividually), no or only minimal increase in test results is observed.

However, if you insist, we can add the aforementioned citation:

Durrani J, Malik F, Ali N, Jafri SIM. To be or not to be a case of heparin resistance. J Community Hosp Intern Med Perspect. 2018;8(3):145-148. Published 2018 Jun 12. doi:10.1080/20009666.2018.1466599

- In the LMWH section, please delete "To the best of our knowledge, there are no official recommendations regarding LMWHs dosage in ECMO" and mention the ELSO guideline.

Actually, in the latest ELSO guideline (2021 ELSO Adult and Pediatric Anticoagulation Guidelines), there is no mention of LMWH so we still believe that the sentence is valid and should not be removed.

- Please describe the current literature evidence of heparin therapy monitoring, apTT vs. anti Xa vs. aPTT + anti Xa

Thank you for this point. The presented paper originally contained an extensive and detailed part on anticoagulation monitoring, which, however, unbearably increased the length of the paper and for this reason, we added only this brief paragraph quoting the ELSO guidelines. Further elaboration on this would require a discussion of individual tests’ use for individual anticoagulants, which would lead to reverting to the extremely long paper. Moreover, the ELSO guidelines we refer to provide this information in greater detail so it would be unnecessary to duplicate this in our paper.

It is, however, likely that your comment originated from our mistake when we have not quoted the ELSO guidelines again at the end of the paragraph and it indeed appeared as if there was a large part of the paragraph unsupported with literature. We have amended this now, thank you for noticing.

- In the table 2, ACT disadvantage column 1st line with typo error, please edit

Thank you for noticing, the typo was corrected.

- In the Bivalirudin section, please delete " To the best of our knowledge, the pleiotropic effect against viruses etc. observed in UFH has not been confirmed in bivalirudin. Limited data suggest that bivalirudin may possess certain anti-inflammatory effects."

We stated this to maintain consistency throughout the paper (this information about antiinflammatory/antiviral action is shown for every discussed anticoagulant). However, we acknowledge that the information „it has no effects“ is unnecessary and we have removed the first sentence. We would, however, prefer to keep the information that limited evidence suggests that bivalirudin may have certain antiinflammatory effects.

- Throughout the manuscript, authors cited COVID-19 literature. I don't think this is needed, given the topic is broadly on anticoagulation in EMCO pts, not pertinent to a specific cohort. So, please delete. 

A lot of the recent ECMO literature dealt with COVID-19 patients because this epidemic caused an unprecedentedly high use of ECMO treatment. For this reason, we believe that this literature (being the most recent) should be kept in the paper, not least because COVID-19 patients still constitute a relatively large proportion of ECMO patients. Many of the findings described in these papers are valid not only for this specific cohort but universally for EMCO treatment. However, we agree that it is unnecessary to highlight (for example) that the antiviral effects of certain anticoagulants were specifically shown in SARS-CoV-2; for this reason, we have removed most of such specific mentions in the text, although the literature remained there for reference if interested readers wished to find out more.